# Pentafluoropropionic Anhydride Derivatization and GC-MS Analysis of Histamine, Agmatine, Putrescine, and Spermidine: Effects of Solvents and Starting Column Temperature

**DOI:** 10.3390/molecules28030939

**Published:** 2023-01-17

**Authors:** Dimitrios Tsikas, Bibiana Beckmann, Svetlana Baskal, Gorig Brunner

**Affiliations:** Core Unit Proteomics, Institute of Toxicology, Hannover Medical School, Carl-Neuberg-Str. 1, 30625 Hannover, Germany

**Keywords:** acylation, amidation, amines, biogenic amines, derivatization, GC-MS, GC column, histamine, pentafluoropropionic anhydride, polyamines, solvent

## Abstract

Gas chromatography–mass spectrometry (GC-MS) is useful for the quantitative determination of the polyamines spermidine (SPD) and putrescine (PUT) and of the biogenic amine agmatine (AGM) in biological samples after derivatization. This GC-MS method involves a two-step extraction with *n*-butanol and hydrochloric acid, derivatization with pentafluoropropionic anhydride (PFPA) in ethyl acetate, and extraction of the pentafluoropropionic (PFP) derivatives by toluene of SPD, PUT, and AGM. We wanted to extend this GC-MS method for the biogenic amine histamine (HA), but we faced serious problems that did not allow reliable quantitative analysis of HA. In the present work, we addressed this issue and investigated the derivatization of HA and the effects of toluene and ethyl acetate, two commonly used water-insoluble organic solvents in GC-MS, and oven temperature program. Derivatization of unlabelled HA (d_0_-HA) and deuterium-labelled HA (d_4_-HA) with PFPA in ethyl acetate (PFPA-EA, 1:4, *v*/*v*; 30 min, 65 °C) resulted in the formation of d_0_-HA-(PFP)_2_ and d_4_-HA-(PFP)_2_ derivatives. d_4_-HA and ^13^C_4_-SPD were used as internal standards for the amines after standardization. Considerable quantitative effects of toluene and ethyl acetate were observed. The starting GC column temperature was also found to influence considerably the GC-MS analysis of HA. Our study shows the simultaneous quantitative analysis of HA as HA-(PFP)_2_, AGM as AGM-(PFP)_3_, PUT as PUT-(PFP)_2_, and SPD as SPD-(PFP)_3_ derivatives requires the use of ethyl acetate for their extraction and injection into the GC-MS apparatus and a starting GC column temperature of 40 °C instead of 70 °C. The PFP derivatives of HA, AGM, PUT, and SPD were found to be stable in ethyl acetate for several hours at room temperature. Analytically satisfactory linearity, precision, and accuracy were observed for HA, AGM, PUT, and SPD in biologically relevant ranges (0 to 700 pmol). The limits of detection of AGM, PUT, and SPD were about two times lower in ethyl acetate compared to toluene (range, 1–22 fmol). The limits of detection were 1670 fmol for d_0_-HA and 557 fmol for d_4_-HA. Despite the improvements achieved in the study for HA, its analysis by GC-MS as a PFP derivative is challenging and less efficient than that of PUT, AGM, and SPD.

## 1. Introduction

Putrescine (PUT), spermidine (SPD), agmatine (AGM), histamine (HA) (Figure 1), and other amines, including cadaverine (CAD) and spermine (SPM), are widely distributed in nature and exert multiple biological functions [1,2]. They are biosynthesized by microorganisms, plants, animals, and humans [3]. *Helicobacter pylori (Hp)* was isolated from subjects infected with the bacterium produced by SPD and HA [4]. Polyamines have therapeutic potential in many diseases, including cancer [5]. Supplementation of SPD has been reported to exert beneficial effects on brain and heart and to extend lifespan [6]. SPD’s autophagy and longevity and other potentially involved mechanisms and functions have been recently reviewed [7,8].

For the measurement of SPD, PUT, and AGM in human serum, we developed a gas chromatography-mass spectrometry (GC-MS) method [9]. This approach includes a two-step extraction with *n*-butanol from alkalinized serum and back to hydrochloric acid, one-step derivatization with pentafluoropropionic anhydride (PFPA) in ethyl acetate, extraction of the pentafluoropropionyl (PFP) derivatives with toluene and GC-MS analysis using a GC column temperature program starting at 70 °C. SPD and AGM form tri-PFP derivatives, i.e., SPD-(PFP)_3_ and AGM-(PFP)_3_, respectively; PUT forms a di-PFP derivative, i.e., PUT-(PFP)_2_ [9]. These observations suggest all amine groups of SPD, AGM, and PUT react with PFPA to form the corresponding PFP amides (Figure 1). In this method, we used a commercially available ^13^C_4_-spermidine (^13^C_4_-SPD) as the internal standard for SPD, PUT, and AGM. By this GC-MS method, we faced difficulties in analyzing reliably HA as a PFP derivative. These included varying derivatization and extraction yields using different organic solvents, notably toluene. Others found by direct inlet probe MS analysis in the electron ionization (EI) mode that the derivatization of HA with PFPA in acetonitrile (30 min, 60 °C) forms a mono-PFP derivative [10]. The derivatization of HA with PFPA in ethyl acetate-toluene-acetonitrile mixture (30 min, 50 °C) and its analysis by GC with flame ionization detection (FID) have been reported, but the structure of the derivative has not been reported [11].

In previous work, we found the organic solvent, toluene, which is commonly used by our and other groups in GC-MS analyses of structurally related pentafluorobenzyl (PFB) derivatives of various endogenous analytes, is associated with considerable carryover effects for some analytes. This was especially the case for the PFB derivative of inorganic nitrate (i.e., PFB-ONO_2_). This phenomenon occurred when we changed the single quadrupole GC-MS apparatus model DSQ by the single quadrupole GC-MS apparatus model ISQ from the same manufacturer (ThermoFisher) [12]. We solved this problem by using ethyl acetate (EA) instead of toluene (TOL) for the extraction of PFB-ONO_2_ from human plasma samples after derivatization of nitrate with PFB bromide [12]. In the present work, we investigated the effects of these two solvents, as well as of two starting temperatures of the gas chromatographic (GC) column, i.e., 40 °C and 70 °C, on the GC-MS analysis of HA, SPD, PUT, and AGM using otherwise identical conditions.

We report here for the first time the derivatization of unlabelled HA (d_0_-HA) and deuterium-labelled HA (d_4_-HA) with PFPA, the structural characterization of the derivative and the quantitative GC-MS analysis of HA. Our present results show previously used derivatization and GC-MS conditions for SPD, PUT, and AGM are optimum for the simultaneous derivatization of HA, AGM, PUT, and SPD by GC-MS only when EA was used for the extraction of the PFP derivatives and when the starting GC column temperature was 40 °C. Commercially available ^13^C_4_-SPD and ^2^H_4_-HA were tested and found to be useful as internal standards for the analytes after standardization.

## 2. Results

### 2.1. Derivatization of Histamine with PFPA and Structural Characterization of its PFP Derivative

The reaction of HA with PFPA resulted in a single intense GC peak in the relevant retention time window of 7 to 14 min. The GC-MS mass spectra of the PFP derivatives of unlabelled HA (d_0_-HA) and ^2^H_4_-HA (d_4_-HA) from EA extracts are shown in Figure 1.

The most intense ions in the GC-MS mass spectra of the PFP derivatives of d_0_-HA and d_4_-HA are *m/z* 256 and *m/z* 260, respectively. Much less intense ions in these mass spectra are *m/z* 279 and *m/z* 283, respectively. The difference of 4 Da in these two ion pairs and the shorter retention time of the d_4_-HA derivative compared to the d_0_-HA derivative are due to the presence of four deuterium atoms in the d_4_-HA derivative. The occurrence of *m/z* 279 and *m/z* 283 in the GC-MS spectra suggests the derivatives each contain two PFP residues. Under the derivatization conditions (30 min, 65 °C) d_0_-HA and d_4_-HA react with PFPA form di-PFP derivatives, i.e., d_0_-HA-(PFP)_2_ and d_4_-HA-(PFP)_2_, respectively. The mass fragments, *m/z* 279 and *m/z* 283, are likely to result from the loss of a PFP residue (CF_3_CF_2_CO, 147 Da) from d_0_-HA-(PFP)_2_ (molecular mass, 403) and d_4_-HA-(PFP)_2_ (molecular mass, 407), respectively. The missing consecutive neutral losses of HF (20 Da) from *m/z* 279 and *m/z* 283 in the mass spectra to generate ions with *m/z* 259 and *m/z* 263, respectively, suggest the PFP residue (147 Da) of the imidazole ring of HA is lost during NICI [9] (Figure 1).

Under the same derivatization, extraction, and GC-MS conditions, the mass spectrum of the PUT derivative contained the most intense ion at *m/z* 340, suggesting formation of PUT-(PFP)_2_. The most intense ion in the mass spectrum of AGM derivative was *m/z* 528 due to AGM-(PFP)_3_. The most intense ions in the GC-MS spectra of the derivatives of SPD (^13^C_0_-SPD) and ^13^C_4_-SPD were *m/z* 361 and *m/z* 365, respectively, suggesting formation of SPD-(PFP)_3_ (molecular mass, 583). ^13^C_0_-SPD-(PFP)_3_ and ^13^C_4_-SPD-(PFP)_3_ eluted virtually at the same time. The difference of 4 Da is due to the presence of four ^13^C atoms in the ^13^C_4_-SPD derivative. These observations confirm previous results for SPD, PUT and AGM [9].

### 2.2. Linearity and Standardization of ^2^H_4_-Histamine

Aqueous solutions of d_0_-HA and d_4_-HA were used for the standardization of the commercially available d_4_-HA. Varying amounts of d_0_-HA (0, 70, 280, 420, 700 pmol) were spiked with a fixed amount of d_4_-HA (nominal amount, 300 pmol). After solvent evaporation to dryness by a stream of nitrogen gas, PFPA derivatization (30 min, 65 °C) was carried out, derivatives were extracted with EA, and GC-MS analysis in the SIM mode (*m/z* 256 and *m/z* 260) was performed. Linear regression analysis between the PAR of *m/z* 256 to *m/z* 260 (*y*) against the amount of d_0_-HA (*x*) resulted in the regression equation, *y* = 0.026 + 0.0011 × *x* (*r*^2^ = 0.9977, *p* < 0.0001). The reciprocal slope of the regression equation corresponds to the amount of d_4_-HA used in this experiment and is calculated to be 906 pmol. In quantitative analyses, the corrected concentration of the internal standard d_4_-HA in its stock solution was used. The PAR of *m/z* 256 to *m/z* 260 in the d_4_-HA sample was determined to be 0.022, indicating an isotopic purity of about 98% ^2^H, which is closed to the declared isotopic purity.

### 2.3. Effects of Ethyl Acetate on the GC-MS Analysis of the Amines as PFP Derivatives

EA extracts of derivatized mixtures of amines were analyzed by GC-MS. The retention time (and the coefficient of variation CV, %) of the PUT derivative was 7.942 min (0.05%). The retention time of the AGM derivative was 9.26 min (0.00%). The retention times of the PFP derivatives of d_0_-HA and d_4_-HA were 10.93 min (0.08%) and 10.90 min (0.07%), respectively. The retention times of the PFP derivatives of ^13^C_0_-SPD and ^13^C_4_-SPD were 11.42 min (0.04%) and 11.42 min (0.00%), respectively.

Plotting the PAR (*y*) of the peak areas of the PFP derivatives of the amines to the peak areas of ^13^C_4_-SPD against the amount of the derivatized amines (*x*) resulted in the following regression equations using EA and OTP40: *y* = 0.66 + 0.023 *x* (*r*^2^ = 0.9925, *p* < 0.0001) for PUT; *y* = −0.27 + 0.029 × *x* (*r*^2^ = 0.9968, *p* < 0.0001) for AGM; *y* = 0.03 + 0.0027 × *x* (*r*^2^ = 0.9893, *p* < 0.0001) for SPD; and *y* = −0.01 + 0.00074 × *x* (*r*^2^ = 0.8949, *p* = 0.0043) for HA.

Plotting the PAR (*y*) of the peak area of the PFP derivatives of the amines to the peak area of d_4_-HA against the amount of the derivatized amines (*x*) resulted in the following linear regression equations using EA and OTP40: *y* = 0.27 + 0.067 × *x* (*r*^2^ = 0.9863, *p* = 0.0007) for PUT; *y* = −1.65 + 0.079 × *x* (*r*^2^ = 0.9737, *p* = 0.0018) for AGM; *y* = 0.18 + 0.0007 × *x* (*r*^2^ = 0.771, *p* = 0.020) for SPD; and *y* = −0.27 + 0.029 × *x* (*r*^2^ = 0.9968, *p* < 0.0001) for HA.

These results suggest PUT, AGM, SPD, and HA can be quantified simultaneously by GC-MS as their PFP derivatives in EA extracts using either ^13^C_4_-SPD or d_4_-HA as internal standards. Yet, the highest linearity is observed when ^13^C_4_-SPD was used as the internal standard for SPD and d_4_-HA as the internal standard for HA.

### 2.4. Effect of Derivatization Time and Long-Term Stability of the Histamine PFP Derivative

Standard curves of d_0_-HA and d_4_-HA were prepared as described above, yet for two derivatization times, i.e., 30 min and 60 min, in duplicate for each concentration. The PFPA derivatization temperature of 65 °C and a reaction time of 30 min were found sufficient for the GC-MS analysis of many amino acids and amines [7,9,12]. Higher derivatization temperatures were not tested in the present study. The samples were placed in the autosampler (at about 19 °C), and the EA extracts were analyzed consecutively in five cycles, beginning from the lower to the higher d_0_-HA amounts. Samples were injected every 20 min. The results of this experiment are illustrated in Figure 2 and indicate close similarity except of the highest amount d_0_-HA that was derivatized for 60 min. The mean CV values for all analyses were 91% for 0 pmol d_0_-HA, 10.7% for 70 pmol d_0_-HA, 9.7% for 280 pmol d_0_-HA, 7.3% for 420 pmol d_0_-HA, and 13.5% for 700 pmol d_0_-HA. Linear regression analysis between the PAR of *m/z* 256 to *m/z* 260 (*y*) and the amount of d_0_-HA (*x*, pmol) resulted in the following regression equations (*r*^2^ range, 0.88 to 0.96, *p* < 0.0001): *y* = −0.2 + 0.0039 × *x*; *y* = −1.3 + 0.0049 × *x*; *y* = −2 + 0.0044 × *x*; *y* = −3 + 0.0048 × *x*; *y* = −3.60 + 0.0042 × *x*; *y* = −6 + 0.0054 × *x*; *y* = −6 + 0.0054 × *x*; *y* = −5 + 0.0043 × *x*; *y* = −7 + 0.0049 × *x*; *y* = −6 + 0.0035 × *x*; and *y* = −10 + 0.0054 × *x*, respectively (Figure 2).

### 2.5. Effects of Ethyl Acetate and Toluene and of the Starting GC Column Temperature on the Amine Analysis by GC-MS as PFP Derivatives

Typical GC-MS chromatograms from analyses of the PFP derivatives in EA and TOL extracts using OTP40 are shown in Figure 3. The PFP derivatives of PUT, AGM, SPD, and ^13^C_4_-SPD had almost identical retention times in EA and TOL extracts. The PFP derivatives of d_0_-HA and d_4_-HA were found in larger abundance in the EA extracts. On a molar basis, the PFP derivatives of PUT and AGM had similar peak areas in both extracts. The PFP derivatives of SPD and ^13^C_4_-SPD had similar peak areas in both extracts as well, but they were considerably smaller compared to those of PUT and AGM. The peak areas of the PFP derivatives of d_0_-HA and d_4_-HA were small in EA and not present in TOL extracts (Figure 3). The main results are reported in Table 1 for the individual amines and include retention time (*t*_R_) and peak area (PA) of the derivatives, linearity between PA (*y*) and amine amount *(x),* and signal-to-noise (*S/N*) values. Table 1 also reports the peak area ratio (PAR) of the PFP derivatives of ^13^C_0_-SPD-(PFP)_3_ and ^13^C_4_-SPD-(PFP)_3_. The detailed results of this experiment for PUT, AGM, and SPD are presented in the Appendix A to this work.

The results of Table 1 suggest PUT, AGM, and SPD can be analyzed with comparable sensitivity using EA or TOL for extraction and OTP40 or OTP70 for GC separation. The mean of the slope of the regression equation of the PAR of ^13^C_0_-SPD/^13^C_4_-SPD is 0.002755 (CV, 4.7%). The reciprocal of the mean value of the slope corresponds to 363 pmol ^13^C_4_-SPD (nominal amount, 300 pmol for standardized ^13^C_4_-SPD).

The results of these experiments with the pure amines were confirmed after extraction of the above-mentioned amines from aqueous buffer of pH 7.4 using a standard two-step *n*-butanol/HCl extraction [9], derivatization with PFPA (30 min, 65 °C), extraction of the derivatives with EA, and GC-MS analysis using OTP40, as described in this work (Table 2). d_0_-HA and d_4_-HA were not detectable by GC-MS when TOL was used for the extraction of the native amines and of the PFP derivatives. d_4_-HA and ^13^C_4_-SPD were used at constant amounts. The PA values of the PFP derivatives of d_4_-HA and ^13^C_4_-SPD in the EA extracts varied by 5.9% and 18.3%, respectively.

### 2.6. Limits of Detection

The peaks shown in Figure 3 were obtained by injecting 1670 fmol of each amine assuming complete derivatization and extraction. The *S/N* values obtained from these analyses are summarized in Table 1. The *S/N* values of PUT were 1087:1 in EA and 685:1 in TOL. The *S/N* values of AGM were 4696:1 in EA and 2573:1 in TOL. The *S/N* values of SPD were 480:1 in EA and 228:1 in TOL. The *S/N* values of ^13^C_4_-SPD were 398:1 in EA and 237:1 in TOL.

The lowest *S/N* values were observed for ^2^H_0_-HA (4:1) and ^2^H_4_-HA (9:1) using EA and OTP40. The lower limit of detection (LOD) values were approximated using *S/N* values of 3:1 (Table 3); see also Table 1. The approximated LOD values were 1670 fmol ^2^H_0_-HA and 557 fmol for ^2^H_4_-HA (in EA). The approximated LOD values were 4.6 fmol (in EA) and 7.3 fmol (in TOL) for PUT; 1.1 fmol (in EA) and 1.9 fmol (in TOL) for AGM; 10.4 fmol (in EA) and 22.0 fmol (in TOL) for ^13^C_0_-SPD; 12.6 fmol (in EA) and 21.1 fmol (in TOL) for ^13^C_4_-SPD.

## 3. Discussion

Histamine (2-(1*H*-imidazol-4-yl)ethanamine, C_5_H_9_N_3_, 111.1 g/mol; p*K*_a_, 6.04 (imidazole NH); p*K*_a_, 9.75, terminal NH_2_) (Figure 1) is a biogenic amine produced from L-histidine by enzymatic decarboxylation. Enzymatic decarboxylation of L-arginine leads to agmatine (AGM). Analogous, enzymatic decarboxylation of L-ornithine generates putrescine (PUT), which is further metabolized to spermidine (SPD) and spermine (SPM).

Many different analytical methods have been reported for the measurement of HA and other amines in biological fluids and tissues [13,14,15]. Reported GC-based methods for HA include its derivatization with various derivatization reagents, such as PFPA [9,10,11] and pentafluorobenzyl bromide (PFB-Br) [4,16]. PFPA, other perfluorated substances [17], and PFB-Br [18] are versatile derivatization reagents in GC-MS. For the measurement of SPD in human serum, we developed a GC-MS method [9]. We faced considerable difficulties in analyzing reliably and sensitively HA as a PFP derivative by the previously reported GC-MS method which uses toluene (TOL) as the extraction solvent [9]. The present work shows these difficulties can be overcome by using ethyl acetate (EA) as the extraction and injection solvent in combination with a low starting oven temperature of 40 °C. This is the first demonstration of the derivatization of HA with PFPA to form a HA-(PFP)_2_ derivative. Use of TOL for the extraction of PFP derivatives is likely to be misinterpreted as the inability of PFPA to react with HA under standard derivatization conditions (e.g., 30 min at 65 °C).

Base-catalyzed derivatization of HA with PFB-Br in anhydrous acetonitrile results in formation of a tri-PFB derivative, suggesting alkylation of the imidazole NH group and two-fold alkylation of the aliphatic NH_2_ group: HA-(PFB)_3_ [4,16]. In the present work, we did not find formation of a tri-PFP derivative of HA using PFPA in EA (PFPA-EA, 1:4, *v*/*v*). Our results suggest under the derivatization conditions used in the present study, i.e., derivatization of HA with PFPA at 65 °C for 30 min or 60 min, results in formation of a di-PFP derivative, i.e., HA-(PFP)_2_, most likely by acylation of the ring imidazole NH group and mono-acylation of the non-ring aliphatic NH_2_ group. Under similar derivatization conditions, PUT, AGM, and SPD react with PFPA in EA to form PUT-(PFP)_2_, AGM-(PFP)_3_, and SPD-(PFP)_3_, respectively. The polyamine cadaverine (CAD, pentane-1,5-diamine; C_5_H_14_N_2_) forms a di-PFP derivative (CAD-(PFP)_2_) which elutes at 8.77 min (OTP40), i.e., behind the PUT-(PFP)_2_ derivative and in front of the AGM-(PFP)_3_ derivative. Obviously, derivatization of HA with PFPA differs from that of PUT, AGM, SPD, and CAD. Other biogenic amines, such as tryptamine and beta-phenyl ethylamine, are expected to be derivatized with PFP, but they are unlikely to interfere with the GC-MS analysis of HA as a PFP derivative.

For the GC-MS analysis of urinary HA, ethyl acetate (EA) was used for the extraction of HA-(PFB)_3_ [16]. In previous work, we used this GC-MS method to identify HA isolated from cultures of *H. pylori* patients [4]. In our group, we commonly use toluene (TOL) for extraction of PFB and PFP derivatives and their subsequent analysis by GC-MS. TOL is entirely immiscible with water and does not require additional treatment with anhydrous Na_2_SO_4_ to remove the remaining water prior to GC-MA analysis. In addition, PFB and PFP derivatives in TOL are less susceptible to hydrolysis than in EA. For instance, PFP derivatives of methyl esters of amino acids are stable for long periods in TOL extracts [19]. On the other hand, we observed GC-MS analyses of PFB derivatives, such as that of nitrate (PFB-ONO_2_), are associated with considerable carryover effects; we found changing the solvent from TOL to EA eliminates this problem and enables reliable analysis of derivatized nitrate [12]. For this reason, we investigated whether water-immiscible GC-compatible organic solvents, such as TOL and EA, may influence the GC-MS analysis of HA. To investigate potential effects of the oven temperature, we tested two GC oven temperature programs with different starting temperatures, i.e., 40 °C and 70 °C, yet with the same oven two-step temperature gradient.

To the best of our knowledge, the present study is the first to report that HA reacts with PFPA to form HA-(PFP)_2_. The results of the present study strongly indicate TOL is not useful for the GC-MS analysis of HA after derivatization with PFPA, although PFP derivatives of PUT, AGM, and SPD can be analyzed even more sensitively in TOL extracts. Changing TOL by EA enables HA analysis as a PFP derivative alongside PUT, AGM, and SPD analysis. The starting GC oven temperature also has an effect, which is considerable for PUT, presumably because of the comparably small molecule and short retention time of its PFP derivative. Among the investigated amines, HA analysis by GC-MS as a PFP derivative is considerably less efficient than the GC-MS analysis of PUT, AGM, and SPD. The PFP derivatives of PUT and AGM are best suitable for sensitive GC-MS analysis. The main reason for the better utility of EA in the GC-MS analysis of HA as a PFP derivative compared to TOL could better solubility and stability of the PFP derivatives in EA. It seems these factors also apply to SPD, albeit to a lower degree. The disadvantage of the use of EA is the requirement of an additional experimental step to remove the remaining water, for instance by using anhydrous Na_2_SO_4_. Yet, this is considered neglectable and could be dispensed by using stronger centrifugation forces of the EA extract.

Spermine (SPM, C_10_H_26_N_4_) is considerably longer than SPD (C_7_H_19_N_3_) and forms a tetra-PFP derivative (SPM-(PFP)_4_) upon derivatization with PFPA under the same derivatization conditions. GC-MS analysis of derivatized SPM showed decomposition, yet not to PUT and SPD [9]. Increasing length of the polyamine PFP derivatives seems to be associated with increasing thermal instability analogous to the methyl ester PFP derivatives of di- and tripeptides, such as glutathione and ophthalmic acid [20,21]. The retention time of the PFP derivatives of the investigated amines is proportional to the molecular weight of the native and derivatized amines (Figure 4), but it is independent of the molar ratio of derivatized to non-derivatized amines (4.01 ± 0.28; CV, 7%). Figure 4 shows AGM and HA behave differently than the other amines, presumably due their guanidine and imidazole groups, respectively. It seems the PFP residue of the guanidine group of AGM increases the volatility of AGM-(PFP)_3_ stronger than the PFP residue of the imidazole group of HA-(PFP)_2_. Other polyamines, such as cadaverine and spermine (Figure 4), and other biogenic amines, such as tryptamine and beta-phenylethylamine (not tested in the present study), are likely not to interfere with the analysis of PUT, AGM, HA, and SPD as their PFP derivatives. They are expected to form PFP derivatives with distinctly different mass spectra and retention times under the same derivatization and GC-MS conditions.

## 4. Materials and Methods

### 4.1. Chemicals and Materials

^13^C_4_-spermidine (^13^C_4_-SPD, 99% ^13^C) hydrochloride was obtained from Biomol (Hamburg, Germany) and was standardized as reported elsewhere [9]. It was tested as the internal standard for SPD, PUT, AGM, and HA. [α,α,β,β-^2^H_4_] Histamine dihydrochloride (98% ^2^H, d_4_-HA) and ethyl acetate were obtained from Merck (Darmstadt, Germany). Toluene and the hydrochloride salts of the unlabelled amines (chemical purity, 97%) were purchased from Sigma (Deisenhofen, Germany). Pentafluoropropionic anhydride (PFPA) was obtained from ThermoScientific (Dreieich, Germany) and used as the derivatization reagent for the amines. Glassware (1.8-mL autosampler vials and micro inserts) was purchased from Macherey-Nagel (Düren, Germany) and used in GC-MS analyses. Aqueous solutions of unlabelled and stable-isotope labelled amines were prepared in 10 mM HCl, diluted with distilled water as appropriate, and stored at −20 °C.

*Safety Considerations*. PFPA is corrosive and malodorous. Inhalation and contact with the skin and eyes should be avoided. All work should be and was performed in a well-ventilated fume hood.

### 4.2. Derivatization Procedure

Aliquots of aqueous solutions of the amines were evaporated to complete dryness by means of a stream of nitrogen gas. The solid residues were treated each with 100-µL aliquots of a freshly prepared PFPA reagent in ethyl acetate (EA), i.e., PFPA-EA (1:4, *v*/*v*). The glass vials were tightly sealed, and the samples were heated for 30 min (in some experiments for 60 min) at 65 °C. Then, solvents and reagents were evaporated to dryness using a stream of nitrogen gas to completely remove remaining solvents, remaining PFPA, and other volatile species. The residues were reconstituted in 200-µL aliquots of pure EA or 200-µL aliquots of pure toluene (TOL) by vortexing for 2 min. After sample centrifugation (5 min, 3750× *g*, 4 °C), each 180-µL aliquots of the EA and TOL supernatants were transferred into micro inserts placed in 1.8-mL autosampler vials. Aliquots (1 µL) of the EA and TOL extracts were injected into the GC-MS apparatus in the splitless mode.

### 4.3. Investigations of the Effects of Ethyl Acetate and Toluene and of the Starting GC Column Temperature on GC-MS Amine Analysis

The effects of EA and TOL and of the starting GC column temperature on amine analysis were investigated using aqueous solutions of the amines as follows. Two separate 10-µL aliquots of an aqueous 30-µM solution of ^13^C_4_-SPD (equivalent to a fixed nominal amount of 300 pmol ^13^C_4_-SPD) were introduced into glass vials. To these samples, 10-µL aliquots of an aqueous 50-µM solution of d_4_-HA (equivalent to a fixed nominal amount of 500 pmol d_4_-HA) were added. Subsequently, separate aliquots (5 µL, 10 µL, 15 µL, 20 µL, and 25 µL) of a 12-µM solution of HA and separate aliquots (5 µL, 10 µL, 15 µL, 20 µL, and 25 µL) of a mixture of SPD, PUT, and AGM containing these amines at 12 µM each were added and mixed by vortexing. The corresponding added amounts were 60 pmol, 120 pmol, 180 pmol, 240 pmol, and 300 pmol for each amine. The samples were derivatized with PFPA-EA for 30 min at 65 °C, and the PFP derivatives were extracted either with EA or TOL, as described above. GC-MS analyses of the extracts were performed by selected-ion-monitoring (SIM) with a dwell-time of 100 ms for each ion. The selected ions used in SIM were *m*/*z* 256 for unlabelled HA (d_0_-HA) and *m*/*z* 260 for d_4_-HA (see Section 2), *m*/*z* 340 for PUT (d_0_-PUT), *m*/*z* 528 for AGM (d_0_-AGM), *m*/*z* 361 for unlabelled SPD (^13^C_0_-SPD), and *m*/*z* 365 for ^13^C_4_-SPD [9]. The electron multiplier voltage was 1900 V. Each sample was injected two times in the order of increasing amounts of the derivatized amines. Analyses were performed in two runs, first with OTP40 and then with OTP70 (see below).

### 4.4. GC-MS Conditions

GC-MS analyses of derivatized amines were performed on a single quadrupole mass spectrometer model ISQ directly interfaced with a Trace 1310 series gas chromatograph (model, manufacturer, city, (state abbreviation if USA and Canada), country) equipped with an autosampler AS 1310 from ThermoFisher (Dreieich, Germany). The GC-MS conditions were similar to those previously reported for PUT and SPD [9]. In brief, the gas chromatograph was equipped with a 15 m long fused-silica capillary column model Optima 17 (0.25 mm I.D., 0.25-µm film thickness) from Macherey-Nagel (Düren, Germany). The injector temperature was kept constant at 280 °C. Interface and ion-source temperatures were set to 300 °C and 250 °C, respectively. Electron energy was 70 eV and electron current 50 µA. Methane was used as the reagent gas for negative-ion chemical ionization (NICI) at a constant flow rate of 2.4 mL/min. Helium was used as the carrier gas at a constant flow rate of 1.0 mL/min.

Two oven temperature programs (OTP) were used that differed in the start temperature only. For OTP40, the starting oven temperature was 40 °C, was held at this temperature for 0.5 min, and then ramped first to 210 °C at a rate of 15 °C/min and finally to 320 °C at a rate 35 °C/min. For OTP70, the starting oven temperature was 70 °C, was held at this temperature for 0.5 min, and then ramped first to 210 °C at a rate of 15 °C/min and finally to 320 °C at a rate 35 °C/min.

GC-MS NICI spectra were obtained in the scanning mode with a rate of 1 s/scan. Quantitative analyses were performed in the SIM mode with a dwell time of 100 ms for each ion. Each sample was injected two times in the order of increasing amounts of the derivatized amines. Analyses were performed in two runs, first with OTP40 and then with OTP70. The 10-µL Hamilton needle of the autosampler was cleaned automatically three times with EA (for EA extracts) or TOL (for TOL extracts (each 5 µL) after each injection. Peak area (PA) values and signal-to-noise (*S/N*) ratios were calculated automatically by the GC-MS software (Xcalibur and Quan Browser). In some analyses, peak area ratios (PAR) of unlabelled to stable isotope-labelled amines were calculated.

### 4.5. Data Analysis and Presentation

Data analyses were performed using GraphPad Prism 7 for Windows (GraphPad Software, San Diego, CA, USA). Chemical structures were drawn using ChemDraw 15.0 Professional (PerkinElmer, Germany). Data analysis in more detail is reported in Section 2.

## 5. Conclusions

The derivatization (30 min, 65 °C) of HA with PFPA in EA (PFPA-EA, 1:4, *v*/*v*) yields a HA-(PFP)_2_ derivative. PFPA acylates the primary aliphatic amine group of HA and the ring imidazole imine group of HA. Under these conditions, the amines, PUT, AGM, and SPD, react with PFPA to form PUT-(PFP)_2_, AGM-(PFP)_3_, and SPD-(PFP)_3_ derivatives, respectively. Simultaneous quantitative analysis of HA, PUT, AGM, and SPD by GC-MS as PFP derivatives is possible when using EA as the extraction and injection solvent for the derivatives and a starting GC column temperature of 40 °C followed by a two-step oven temperature gradient. d_4_-HA is useful as an internal standard for HA. TOL allows simultaneous analysis of PUT, AGM, and SPD but not alongside HA. The HA-(PFP)_2_ derivative is stable in EA extracts for at least 30 h at room temperature. Despite the analytical improvements achieved for HA, its analysis by GC-MS as PFP derivative is still challenging and less efficient than that of PUT, AGM, and SPD.

## Data Availability

The study did not report any data.

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
