# Peer review of "Pentafluoropropionic Anhydride Derivatization and GC-MS Analysis of Histamine, Agmatine, Putrescine, and Spermidine: Effects of Solvents and Starting Column Temperature"

_molecules, 2023, doi:10.3390/molecules28030939_

Round 1

Reviewer 1 Report

This is an interesting and novel article, for the simultaneous derivatization and GC-MS determination of Histamine, Agmatine, Putrescine and Spermidine with pentafluoropropionic anhydride (PFPA) in ethyl acetate.

11)     Line 58-59 “By this GC-MS method, we faced difficulties in analyzing reliably HA as PFP derivative”. Please mention in a few words these difficulties for the reader to understand the usefulness of the new method.

22)     Line 68-Scheme 1 caption: The MW is for the derivatized molecule and not for the starting chemical substance but is not clear for the reader. Please clarify this. Additionally, a larger fond for the MW of each derivatized molecule would help the reader to understand.

33)     Scheme 1: Is there any possible mechanism to explain the formation of bonds between the nitrogen of these compounds and the carbon of pentafluoropropionic anhydride?

44)     Line 93: There is something missing “[            -2H4]Histamine dihydrochloride”. Please correct this.

55)     Line 102: Pentafluoropropionic anhydride can cause severe skin burns and eye damage. Where there any other protection measures (gloves, glasses etc) taken for the safety of the analyst?

66)     Line 109: Why the glass vials were heated at 65 OC and then evaporated to dryness? Why did not the authors evaporate to dryness in the first place? It is not clear why the heating step is necessary for the derivatization procedure.  

77)     Line 242:Why the authors picked 30 and 60 mins as derivatization times? Is it possible to improve the derivatization process for 90 mins or did the authors conclude that complete derivatization is accomplished at 60 mins?

88)     Did the authors try higher derivatization temperatures? There are published works that refer to derivatization of amines with pentafluoropropionic anhydride at 100 oC? Is it possible with higher temperatures to improve the effectiveness of the method?

99)     Line 329-338: The comparison of the proposed method with the two-step n-butanol/HCl extraction method is rather confusing. A comparative table may clear this issue.  

110)  Line 340-349: Also, a table would be more helpful for the reader to understand the limits of detection of this method.

111)  Additionally, the report of so many numbers in the text is diminishing the reader's interest especially in paragraphs 3.3 to 3.6. A restatement of these paragraphs using tables or diagrams where possible would assist the reader. Additionally in the regression equations (e.g., lines 228-235) why are there no errors in both the slope and the intercept?  

112)  Please keep the reference in the same format. For example, in line 507 the issue of the journal is in bold and not in italics as the other ones. Additionally, why there are DOI numbers only for a few papers? Only these references have DOI numbers?

Author Response

siehe attachment

Reviewer 2 Report

The manuscript with reference number molecules-2120521 entitled " Simultaneous Pentafluoropropionic Anhydride Derivatization and GC-MS Measurement of Histamine, Agmatine, Putrescine and Spermidine: Effects of Solvents and Starting Column Temperature" reports simultaneous quantitative analysis of HA, PUT, AGM and SPD by GC-MS as PFP derivatives when using EA as the extraction under mild reaction conditions. The result of research shows that HA, AGM, PUT and SPD in biologically relevant ranges (0 to 700 pmol). The limits of detection were 1670 fmol for d0-HA and 557 fmol for d4-HA. However, minor revisions should be made before the final acceptance of this paper for publication in Molecules.

1.      How should the author calculate the corresponding peaks in the mass spectrum and indicate whether they are molecular ion peaks or additive ion peaks or something else.

2.      Whether the retention time of other bioamines such as tryptamine, beta-phenylethylamine interferes with the determination of Histamine, Agmatine, Putrescine and Spermidine.

3.      Deviations or errors throughout the article should be given.

4.      There are many grammatical errors and formatting in the manuscript, which are suggested to be corrected.

5.      In the introduction section, a few highly relevant references should be cited and discussed properly: Separation and Purification Technology, 2023, 304, 122342;  Angewandte Chemie International Edition, 2022, 61, e 202207209.

Author Response

siehe attachment

Reviewer 3 Report

The manuscript presented by the authors is interesting, however, not validating the developed method with real samples/matrices makes me doubt about its publication in Molecules. 

Line 93: Missing information in the blank space between brackets.

Line 118: The sentence does not make sense, please rewrite it. 

Line 196: The mass spectra of amines derivatized in toluene are shown in reference 9. derivatized in toluene. Please include those same spectra but in ethyl acetate.

Line 235: If the correlation coefficient for HA is 0.0068, its quantification is not possible. Please check it.

Section 3.5: Present the results in a clearer way. A table could be used to compare the parameters obtained for each amine. 

Line 331 to 338: the significant figures in the confidence limits are wrongly expressed. Please correct them.

Section 3.6: Do the LODs presented correspond to the method or to the technique? On the other hand, it does not make sense to indicate the LODs of the deuterated or 13C-labeled compounds. To complete the characteristics of the analytical method developed you should also indicate the LOQs as well as the intra-day and inter-day reproducibility.

 Line 361 to 368: The statement about the method is very strong when it has not been tested on real matrices/samples. 

You should validate the developed method by the determination of these amines in serum samples, for example since this sample was used in reference 9.

Include a table comparing the method presented in the manuscript with others in the literature, including derivatizing agent, separation conditions, amines, internal standard, reproducibility, LOD, LOQ and application to samples. 

The conclusions section are a summary of the results, and the first sentence does not make sense. Please rewrite them. 

Round 2

Reviewer 1 Report

The authors have now provided a revised version of their manuscript with modification of all points raised by the reviewers, so i believe the manuscript has been sufficiently improved and now it is in and the proper form for its publication in Molecules.

Reviewer 3 Report

Table 2. The significant figures in the confidence limits are still wrongly expressed. Please correct them.

Clarify that the LODs correspond to the technique.

Deuterated internal standar are used to quantify the corresponding not deuterated analytes and correct the errors throughout the procedure. In addition, these LOD values can confuse the reader since the importance is in the determiantion of non-deuterated compound. Therefore, it makes no sense to calculate the LODs of the deuterated compounds. On the other hand, to know if there is a matrix effect you should perform the determination on a real sample and compare the slopes of an external calibration and a calibration of standard additions. 

I still think that the development of a methodology is of little interest from a scientific point of view if it is not tested on a real sample. The real difficulties are found when the developed method is applied to a real matrix.